# Depth Completion and Super-Resolution with Arbitrary Scale Factors for Indoor Scenes [note 1]

**DOI:** 10.3390/s21144892

**Published:** 2021-07-18

**Authors:** Anh Minh Truong, Wilfried Philips, Peter Veelaert

**Affiliations:** TELIN-IPI, Ghent University—imec, St-Pietersnieuwstraat 41, B-9000 Ghent, Belgium; Wilfried.Philips@UGent.be (W.P.); Peter.Veelaert@ugent.be (P.V.)

**Keywords:** depth super-resolution, depth completion, deep guided filter

## Abstract

Depth sensing has improved rapidly in recent years, which allows for structural information to be utilized in various applications, such as virtual reality, scene and object recognition, view synthesis, and 3D reconstruction. Due to the limitations of the current generation of depth sensors, the resolution of depth maps is often still much lower than the resolution of color images. This hinders applications, such as view synthesis or 3D reconstruction, from providing high-quality results. Therefore, super-resolution, which allows for the upscaling of depth maps while still retaining sharpness, has recently drawn much attention in the deep learning community. However, state-of-the-art deep learning methods are typically designed and trained to handle a fixed set of integer-scale factors. Moreover, the raw depth map collected by the depth sensor usually has many depth data missing or misestimated values along the edges and corners of observed objects. In this work, we propose a novel deep learning network for both depth completion and depth super-resolution with arbitrary scale factors. The experimental results on the Middlebury stereo, NYUv2, and Matterport3D datasets demonstrate that the proposed method can outperform state-of-the-art methods.

## 1. Introduction

In practice, the resolution of a depth map acquired by an affordable depth sensor is lower than that of the corresponding RGB image, due to technological limitations. This limits the accuracy of applications that require depth information obtained by depth sensors. To address this problem, many studies have considered super-resolution for depth maps [1,2,3,4,5,6,7,8,9]. The goal of DSR is to obtain high-resolution depth maps from low-resolution input depth maps. Recent research on DSR [6,7,8,9] has designed and trained specific upscaling modules for a fixed number of integer scaling factors (e.g., ×2, ×4, or ×8). Although deep learning methods can outperform other classical approaches, existing deep learning methods are often designed only for upscaling depth maps with an integer scaling factor. However, arbitrary scaling is important in hybrid camera networks with RGB and depth sensors, where the sensors have different resolutions, such as indoor camera networks for novel view synthesis. For example, consider a camera network with an Intel RealSense R200 (640×480 depth map) and an RGB camera with a resolution of 2048×1536. If the magnifications of the lenses are equal, the scale factor is 3.2, which is not an integer. In this paper, we mainly focus on indoor camera networks, where both depth completion and DSR need to be considered.

In addition to the limited resolution of depth maps collected from depth sensors, depth values around the corners and along the edges of observed objects are usually inaccurate or missing [10]. Depth completion is the process of reconstructing regions with missing depth values based on the depth information of the remaining part of the depth map and the visual information of the color image. Similar to DSR, researchers have intensively studied depth completion over the last decade, in order to improve the quality of depth maps acquired by depth sensors [11,12,13,14,15].

To the best of our knowledge, although methods for depth completion and DSR have been proposed, all previous work has treated depth completion as an issue separate from DSR. This also means that depth completion and DSR cannot be jointly solved. For example, Li et al. [15] proposed a pipeline to obtain reconstructed depth super-resolution images. In this pipeline, depth completion is applied to the result of DSR. Thus, depth completion needs to be processed on high-resolution images, which leads to a high computational cost. Furthermore, upscaling the resolution of the LR depth map also means increasing the size of the missing areas. Thus, the size of the depth completion network must also be larger (i.e., with a larger receptive field), in order to effectively recover the missing areas on the SR depth map; this also increases the computational cost.

In addition, DSR and depth completion require local information of a different nature. DSR requires local information from neighboring pixels in a small area around the pixel, in order to analyze the characteristics of the textures in the low-resolution image. Then, high-quality textures are generated, based on these characteristics, in the super-resolution image. On the other hand, depending on the size of the holes on the depth map, depth completion might require local information from a much larger area around the pixel, in order to derive the value of the area with missing information. Therefore, joint optimization of the DSR and depth completion tasks is also a challenging problem, due to the different nature of these tasks. In our work, we propose a novel method to perform both depth completion and DSR with arbitrary scale factors. We summarize the contributions of the proposed novel approach as follows:In our work, we propose a novel DSR network that reuses the extracted features from the completion network on LR images, in order to minimize the network size and computation time for real-time applications;We upscale low-resolution depth maps with an arbitrary scaling factor, based on the combination of the results of different projection functions. These projection functions are learned to project the pixels in the LR depth map to an SR depth map, based on a DCNN;We propose a deep neural network for depth completion, based on a deep guided filter, in order to produce better reconstructed depth maps than state-of-the-art depth completion methods.

The remainder of this paper is organized as follows. In Section 2, we review the previous studies focused on various topics related to DSR and depth completion. In Section 3, we explain the proposed method for DSR with arbitrary scaling factors and depth completion, as well as the implementation details. We present the experimental setups, evaluation metrics, experimental results, and detailed analysis of our experiments in Section 4. Finally, Section 5 concludes the paper.

## 2. Related Work

**Depth estimation.** Depth estimation is a problem with a long history of research. The goal of depth estimation is to find the corresponding depth of pixels in the scene. The estimated depth can then be used for other purposes, such as in 3D vision applications. Traditionally, depth estimation has been performed based on triangulation and stereo matching in stereo vision systems [16,17]. Alternatively, depth estimation can be accomplished using depth sensors, such as the Kinect sensor [18]. However, accurately estimating depth from just a single RGB image remains a huge challenge [19]. With the development of machine learning algorithms, many studies have shown promising results in recent years [19,20,21,22,23,24,25]. Although estimating depth from a single image is not as accurate and detailed as estimating depth from a depth sensor, these results show that we can apply feature learning methods to extract the spatial features from color images. Furthermore, these features can be used as secondary information, in order to improve the quality and fill in missing depth values of the depth map captured by the depth sensor.

Eigen et al. estimated the depth map from a single image using a multiscale DCNN [20]. Lee et al. generated multiple overlapping depth maps from different parts of a single image using a DCNN based on Fourier domain analysis [22]. Then, they combined these overlapping depth maps to generate the final depth map. In [22], the authors also proposed a depth-balanced Euclidean loss, in order to balance the impact of distant and near objects during the training process. Thus, the network can estimate the depth of both near and far objects equally well. Fu et al. [23] proposed atrous spatial pyramid pooling with multiple large receptive fields through dilated convolutional operations, in order to extract the spatial features of the scene at multiple scales. In [24,25], the lightweight deep network also showed impressive performance, in terms of speed and accuracy, in depth estimation and semantic segmentation. In [25], this method improved the skip connection of the UNet architecture [26], by replacing the concatenation operation with chained residual pooling to better capture contextual information.

Although single-image depth estimation is not our primary goal, the above work elucidates which architectures are suitable for estimating the spatial structure of a scene. In our work, one of the goals is to quickly fill in missing information regions in depth maps captured by depth sensors for real-time applications. Therefore, we also investigated lightweight DCNN architectures to strike a balance between accuracy and speed for depth completion and DSR.

**Image inpainting.** Image inpainting is the process of reconstructing missing areas in an image, which can used for a number of applications such as filling missing areas for novel view interpolation or image editing to remove unwanted objects. Depth completion is a specific case of inpainting. However, the error in depth completion is usually much more serious than the error in RGB inpainting. In image inpainting, the result usually does not need to be numerically accurate, as long as it looks realistic enough to a human. Typically, a mask is also provided to indicate the missing area on an image; however, this information may become faded after passing through a few layers in the DCNN. Liu et al. [27] introduced partial convolution, in order to reinforce the information of the missing area in the DCNN. Mimicking textures from other regions using a contextual attention mechanism has been shown to be effective [28,29].

While some concepts of image inpainting are also very useful for depth completion, image inpainting and depth completion are still very different. For image inpainting, we have only one source of information. As for depth completion, we have the additional information from the color image. Therefore, in many cases, the spatial structure of the missing parts in the depth map can still be derived from the color image, which can be helpful for the depth completion problem. On the other hand, mimicking information from other image regions—which can be quite powerful in image inpainting—is often not a good idea for depth completion. The main goal of depth completion is to estimate correct depth values for the missing areas, rather than simply mimicking the texture from other areas in the image.

**Depth completion.** The goal of depth completion is to fill in missing holes in depth maps, based on the raw depth observations and corresponding RGB images [13]. In Zhang’s work [13], the authors extracted the structural information of the scene geometry, based on the surface normals and occlusion boundaries from the RGB image. Then, the depth values were interpolated, based on the local neighborhood and the extracted structural information. In [30], Huang et al. applied the contextual attention mechanism to modify the interpolation, based on the depth completion method of [13]. Although the method achieved high accuracy, using multiple submodules of the DCNN is computationally expensive. Furthermore, References [13,30] only exploited the guidance information for depth completion based on the predicted surface normals and occlusion boundaries estimated by the DCNN. This means that the error in the prediction of the surface normals and the occlusion boundaries can mislead the depth completion process. In our work, we tried to extract the guidance information directly from color images. This allows the proposed network to use spatial information other than surface normals and occlusion boundaries.

**Depth super-resolution.** DSR is the process of improving the spatial resolution of depth maps. DSR methods can be divided into two main classes: those that use the high-resolution color image as a guide to improve the spatial resolution and those that use only the depth map. Conventional and deep learning methods have been proposed in both classes. Some examples of conventional learning methods are Markov random fields, autoregressive models [1,2,3,5,31], rigid body self-similarity [32], sparse representation, and dictionary methods [4,33,34,35,36,37]. Song et al. [7] proposed one of the early works of DSR based on an end-to-end DCNN. Riegler et al. [8,38] proposed the use of anisotropic total generalized variation regularization in a DCNN to model the piecewise affine structures in depth data. Meanwhile, Hui et al. [39] upscaled from LR depth maps based on deconvolutions (fractionally strided convolutions).

Later, Song et al. [9] proposed considering the DSR task as a series of novel-view synthesis steps, where a pixel in the LR depth map is upscaled into r×r pixels in the SR depth map (*r* is the DSR scale factor). The authors accomplished this by upscaling a pixel from the LR depth map to the SR depth map using r×r projection functions. Each projection function in [9] was implemented by a separate DCNN. Thus, r×r “virtual” depth maps were generated. The final SR depth map was then generated by rearranging the r×r “virtual” depth maps. In particular, the authors implemented four fixed deep networks to synthesize four different “virtual” depth maps. Song et al. then rearranged them to generate an SR depth map (with the scale factor equal to two). Their approach greatly improved the accuracy of DSR, compared to previous works. However, this approach does not work in the case of noninteger numbers, where one pixel in the SR depth map is the combination of multiple pixels in the LR depth map. Furthermore, this approach requires a different set of subnetworks for each integer scaling factor. Despite the impressive accuracy, the authors in [9] considered the upscaling problem as the projection of a pixel in the LR depth map to r×r pixels in the SR depth map. Thus, it is not suitable for noninteger upscale factors, where a pixel in the SR map can be the combination of two or more pixels. In [40], we produced several candidate depth maps using a RDN [41]. Then, these candidate depth maps were projected onto the SR image plane. Finally, the candidates were fused based on the weight matrices generated by the meta upscale module [42]. In [40], we applied a classical image inpainting method for depth completion. The filled depth values were simply interpolated, based on the neighboring pixels. Thus, this approach only works for small missing depth regions.

## 3. Proposed Method

Let $D^{HR}$ denote the HR depth map, $I^{HR}$ denote the HR color image, $D^{LR}$ denote the low-resolution depth map, $I^{LR}$ denote downscaled image (to match the resolution of $D^{LR}$), and $M^{LR}$ denote the mask. Note that $M^{LR}$ can be obtained from the confidence map provided by the depth sensors (by thresholding), in order to indicate pixels without depth values. The architecture of our network is depicted in Figure 1. The network consists of three main modules: A Weight Prediction Module, a Generator Module, and a Depth Completion Module. In our work, the low-resolution input depth map $D^{LR}$ and low-resolution (downscaled) RGB input image $I^{LR}$ are first fed into the Depth Completion Module, in order to extract the deep features and produce the inpainted depth map $D^{C}$. The inpainted depth map is then projected onto the SR image plane. Then, it is modulated by the residual maps (generated by the Generator Module) based on the elementwise summation of two matrices. The deep features extracted at the last convolutional layer of the Depth Completion Module are also fed into the Generator Module. The Generator Module then generates the “virtual” depth maps from these deep features. The Weight Prediction Module generates the weight maps, which are then used to fuse the “virtual” depth maps. Finally, the output is synthesized based on the weighted average of the generated depth values from the “virtual” depth maps.

In this paper, we upscaled the low-resolution depth map $D^{LR}$ with scale factor *r* by combining the different projections of the input LR depth map to the SR image plane. The projections are generated by the RDN in the Generator Module. We considered each channel of the output from the Generator Module as a residual map, which is then combined with the original depth values to generate the possible depth values on the SR depth map. The result of this process is multiple projected depth maps on the SR image plane. Finally, the depth values of the SR depth map are synthesized based on the projected depth maps and the guidance of the weight map (generated at the Weight Prediction Module). Thus, the network can learn to select and combine the SR from the projected depth maps, with respect to the scale factors.

### 3.1. Depth Completion Module

The architecture of the proposed Depth Completion Module is depicted in Figure 2. The module is composed of four multistage DCNN branches: A Depth branch, a Guidance branch, a Mask branch, and a Fusion branch. We denote the color feature map and the guidance feature map by $F_{l}^{CB}$ and $F_{l}^{GB}$, respectively, extracted from the color image by the Guidance branch in the *l* stage. Let $F_{l}^{DB}$ denote the depth feature map output of the Depth branch in the $l^{\mathrm{th}}$ stage, $F_{l}^{MB}$ denote the validity feature map output of the Mask branch in the *l* stage, and $F_{l}^{FB}$ denote the fusion feature map generated by combining the depth feature map and the color feature map in the *l* stage with the fusion feature map in the (l+1) stage.

In our work, we first extracted multiscale depth feature maps from the LR depth map to capture the spatial structures of the scenes, through the process of spatial dimension reduction in the Depth branch (Figure 2). These features are the main source of information used to synthesize the inpainted depth map later in the network. The spatial dimension reduction also reduces the area of large hole regions. This allows the convolutional kernels for pixels located in the middle of large holes to gather information (as deep features) of the nonhole surrounding regions. These features are then used to estimate the depth values of pixels located in hole regions. Although this can be done by applying convolutional layers with sufficient times, downsampling the spatial dimension (by 16 times) in the fifth stage of the depth branch can reduce both the computational cost and the required memory space of the network. However, downscaling the feature maps also degrades fine details, such as edges. Thus, the extracted depth features of previous stages of the depth branch are also retained, in order to synthesize the inpainted depth map.

Even though the extracted multiscale depth features can describe the information from neighboring regions, the network still has very few clues to complete the missing regions from just these features. For example, the network cannot know the shape of an object or where the boundary of an object is. Surface normals and occlusion boundaries estimated by the DCNN from color images were used as guidance information to complete the missing regions in [13,30]. However, as shown in [24,25], the network can also estimate the depth values from just a color image. Thus, in this paper, we attempted to extract the guidance features directly from the color image using the DCNN in the Guidance branch to synthesize the depth values. The depth features extracted from the LR depth map can be combined, based on the guidance information extracted by the Fusion branch, in order to fill in the missing depth areas.

Furthermore, in practice, depth sensors can only sense depth information within a fixed range. In other words, the depth sensors cannot estimate the depth in areas outside the depth range. This creates large areas of missing depth values in the depth map (see, e.g., Figure 3). If a hole area is larger than the receptive field of the DCNN, the DCNN cannot fill in the pixels located deep in the center of this hole area. In this case, the spatial information extracted from the color image provides information to estimate the depth values. For instance, missing edges in the depth image usually coincide with color image edges, and this fact can be used to restrict the depth values to be completed. Thus, we also generate validity maps (through the Mask branch; see Figure 2) to indicate where the features extracted from the color image are more important than those extracted from the depth map. To do this, we first extract the color feature map $F_{l}^{CB}$ using the DCNN in the $l^{\mathrm{th}}$ stage of the Guidance branch. Then, $F_{l}^{GB}$ in the $l^{\mathrm{th}}$ stage of the Guidance branch is computed, based on $F_{l}^{CB}$ and $F_{l}^{MB}$, as follows:(1)$$F_{l}^{GB}=F_{l}^{MB}⊙F_{l}^{CB}\;,$$
where ⊙ denotes elementwise multiplication. This allows the network to amplify the values of the guidance features in necessary regions and reduce the values of unimportant features.

Finally, the inpainted depth map is progressively synthesized in the Fusion branch. In our work, the Fusion branch starts with the combination of the fifth-stage depth feature map and the fifth-stage guidance feature map. Due to the large receptive field at this stage, the network is better able to fill possible large hole regions. The fusion results in the fifth stage are then gradually upscaled and combined with the guidance features from the fourth stage to the first stages. As explained earlier, this helps the network to improve the details that were degraded after the spatial dimension of the depth feature maps was reduced. In this paper, we propose the Fusion Block (Figure 4), in order to fuse the depth features and the guidance features, based on the idea of a trainable guided filter [45]. It receives three different feature maps: $F_{l}^{GB}$, $F_{l+1}^{FB}$, and $F_{l}^{DB}$. In the Fusion Block, both the linear coefficients $A_{l}$ and $B_{l}$ of the deep guided filter are learned from the data through the Conv Block (Figure 4). The Conv Block consists of two different 3×3 convolutional layers with Leaky ReLU activation [46]. The linear coefficient matrix $A_{l}$ is learned, in order to dynamically select depth features for each channel and each spatial location on the depth feature maps. On the other hand, the linear coefficient matrix $B_{l}$ is learned to reinforce the information on the depth feature maps, when needed. As mentioned earlier, to enhance the details of the reconstructed depth map, $F_{l}^{DB}$ and $F_{l+1}^{FB}$ are fused to generate the merged depth feature map $F_{l}^{M}$. We generate the merged depth feature map based on the residual learning $F_{l}^{M}=F_{l+1}^{FB}+F_{l}^{DB}$ [21]. Residual learning has proven its effectiveness to preserve the details of the output image while performing feature map upsampling within the network [21]. Note that, as there is no previously fused feature map in the fifth stage, the first merged depth feature map is computed as $F_{l}^{M}=F_{l}^{DB}$ instead. Then, the guided feature map $F_{l}^{G}$ is computed as follows:(2)$$F_{l}^{G}=A_{l}⊙F_{l}^{M}+B_{l}\;.$$

As modification based on image color can sometimes be misleading (e.g., the edges in a color image do not always coincide with the edges in the corresponding depth map), the output depth maps may contain minor artifacts. Thus, we apply an RDN with 8 layers to $F_{l}^{G}$, in order to obtain a better fused feature map $F_{l}^{FB}$. To produce the inpainted depth map, we apply a single 3×3 convolutional layer with ReLU activation to obtain the final fused feature map $F_{1}^{FB}$. Finally, both the reconstructed depth map and the final fused feature map $F_{1}^{FB}$ are fed into the next stage of the network, in order to produce the SR depth map.

### 3.2. Upscaling with Arbitrary Scale Factors

In [9], Song et al. projected a pixel in an LR depth map to an SR depth map using r×r networks. Their approach was designed for integer scale factors only. In the case of a noninteger scale factor, one might think that applying a good traditional interpolation method, such as bicubic interpolation, to the upscaled depth map (the closest integer scale factor) could accomplish the task. However, our experimental results showed that the quality of the interpolated depth maps was greatly reduced (which will be shown later in Section 4.5). This might occur due to the fact that interpolating float values is much more difficult than integer value interpolation. Furthermore, the upscaled depth maps produced by the network already contain some incorrectly estimated depth values. These errors are very small (e.g., the average RMSE of scale factor 2.0 on the Middlebury dataset was 0.891 on a scale of 255); however, it might have a bad impact on the interpolation results. Thus, the depth value of a pixel in the SR depth map should be interpolated directly from the depth values in the LR depth map. The main idea of our work was to synthesize the depth value of a pixel on an LR depth map, based on a set of multiple learnable projection functions, to handle the arbitrary scale factor. Moreover, we selected only the relevant projection functions, based on both the relative location of the pixel and scale factor, through the learnable weight prediction module. To do so, we first generate multiple “virtual” depth maps using the Generator Module (Figure 1). Then, we selectively combine them with a specific weight vector for each scale and location in the depth map, based on the guidance of the Weight Prediction Module.

Indeed, this process is very similar to the nearest neighbor upscaling method. However, at location (i,j) in the SR depth map, multiple possible depth values (from the “virtual” depth maps) are generated by the Generator Module, rather than just one value at location $\left(i^{\prime }=\lfloor \frac{i}{r}\rfloor ,j^{\prime }=\lfloor \frac{j}{r}\rfloor \right)$ in the completed LR depth map. Then, the proper depth value at location (i,j) in the SR depth map is combined, based on the weight vector produced by the weight prediction module, with regard to the relative location and scale factor *r*. At first glance, it may seem very naive to consider only the information from a single location on the LR depth map. However, the depth values of the “virtual” depth maps generated by the DCNN are slightly modified from the original values, based on the neighboring pixels. Thus, instead of selecting the same depth value from location $\left(i^{\prime },j^{\prime }\right)$ in the LR, the proposed network generates the depth values for different scales and relative locations $L^{relative}\left(i,j,r\right)$ with respect to location (i,j) in the SR depth map.

The Generator Module (upscaling module) first takes the reconstructed depth map $D^{C}$ as the input to DSR. It has been shown, in [21,47], that the RDN can produce better image quality for image generation tasks. Thus, we applied the idea of residual learning based on the RDN [47] to generate multiple residual maps. These residual maps are combined with the original depth values to generate the “virtual” depth maps by the Generator Module. Then, the “virtual” depth maps are projected onto the SR image plane. Let $D_{t,i,j}^{V}$ denote the projected depth value of the $t^{\mathrm{th}}$ “virtual” depth map to the location (i,j) in the SR depth map. Then, $D_{t,i,j}^{V}$ is computed as follows:(3)$$D_{t,i,j}^{V}=D_{\left\lfloor \frac{i}{r}\rfloor ,\lfloor \frac{j}{r}\right\rfloor }^{C}+R_{t,\lfloor \frac{i}{r}\rfloor ,\left\lfloor \frac{j}{r}\right\rfloor }\;,$$
where $R_{t,\lfloor \frac{i}{r}\rfloor ,\left\lfloor \frac{j}{r}\right\rfloor }$ is the residual value at location $\left(\left\lfloor \frac{i}{r}\right\rfloor ,\left\lfloor \frac{j}{r}\right\rfloor \right)$, $D_{\lfloor \frac{i}{r}\rfloor ,\left\lfloor \frac{j}{r}\right\rfloor }^{C}$ is the depth value of the LR depth map at location $R_{t,\lfloor \frac{i}{r}\rfloor ,\left\lfloor \frac{j}{r}\right\rfloor }$, and *r* is the scale factor.

As a result of this process, *T* projected “virtual” depth maps ($D_{0}^{V},D_{1}^{V},\ldots ,D_{T}^{V}$) are generated. Finally, the SR depth map is constructed by combining the projected “virtual” depth maps, with regard to the guidance weight vectors generated by the Weight Prediction Module (see Figure 1):(4)$$D_{i,j}^{SR}=\sum_{t=0}^{T}\left(W_{t,i,j}^{GB}\cdot D_{t,i,j}^{V}\right)\;,$$
where $W_{t,i,j}^{GB}$ is the guidance weight of the $t^{\mathrm{th}}$ projected “virtual” depth map at location (i,j) and · represents the multiplication operator. Instead of using the location of the pixel as an input to the Weight Prediction Module, we applied the same idea of using relative location as [40,42]. We did not use the locations of the pixels, as the number of locations in an image can be very large. Then, for interpolation of the pixel at location (i,j) in the SR depth map, the location of the source information on the LR depth map can be computed as $\left(\lfloor \frac{i}{r}\rfloor +\left\{\frac{i}{r}\right\},\left\lfloor \frac{j}{r}\right\rfloor +\left\{\frac{i}{r}\right\}\right)$. As the integer shift in the integer parts of the location is just a translation, it is enough to consider the fractional part of the location for optimal interpolation. For each depth value at location (i,j) in the SR depth map, we compute the relative location with regard to the scale factor $L^{relative}\left(i,j,r\right)=\left(L(i,r),L(j,r),1/r\right)$ and feed it into the Weight Prediction Module. The Weight Prediction Module then produces the $W_{t,i,j}^{GB}$ at location (i,j), as mentioned earlier. The L(·) function is described as follows:(5)$$L\left(i,r\right)=\frac{i}{r}-\left\lfloor \frac{i}{r}\right\rfloor .$$

By using only the fractional part of the location, not only is the range of values normalized to $\left[0,1\right)$, but the redundant integer shift can also be omitted. This allows the Weight Prediction Module to learn the weights for arbitrary scale factors effectively.

## 4. Experiments and Results

In this work, we implemented three different variants of the proposed network, in order to find out which architecture was the best for extracting spatial features from color images. These variants are named DCSN-RGBM, DCSN-EfficientNet, and DCSN-UNet, respectively. In the first variant, DCSN-RGBM, the Guidance branch was implemented based on EfficientNet, while the Depth branch was implemented based on UNet [26]. We chose EfficientNet as it has shown impressive results on the ImageNet dataset [48]. This variant was implemented not only for DSR, but also depth completion, as the downsampling layers can increase the size of the receptive field of the network. In the second variant, DCSN-EfficientNet, the Guidance branch was also implemented based on EfficientNet, while both the Depth and Mask branches were implemented based on UNet. For the third variant, DCSN-UNet, we replaced EfficientNet in the Guidance branch with a UNet encoder [26]. For the Depth and Fusion branches, we used the encoder and decoder architectures of UNet. For DSR, RDN has been used in most state-of-the-art DSR methods [7,9]. Thus, we also implemented a variant DCSN-RDN of the proposed method by replacing the Depth Completion Module with an RDN with 8 convolutional layers. This version of the network was much smaller than DCSN-RGBM, DCSN-EfficientNet, and DCSN-UNet.

We first evaluated the performance of DCSN-RGBM, DCSN-EfficientNet, and DCSN-UNet, trained for the depth completion task on the Matterport3D dataset [43]. We trained all the variants on the training set and tested them on the testing set. We show the comparison of these variants, along with state-of-the-art methods, in Section 4.4.

To evaluate the DSR performance of the proposed method, we selected the Middlebury Stereo and NYUv2 Depth datasets [17,49,50,51,52,53]. As DCSN-EfficientNet achieved the best performance for the depth completion task, we only evaluated this variant in the DSR task. Although the Middlebury stereo dataset has been widely used to evaluate the performance of DSR methods, this dataset was not captured by depth sensors. Therefore, the depth maps in the Middlebury stereo dataset do not have the same characteristics as the depth maps captured by real depth sensors (e.g., noise and edges). Thus, we evaluated the DSR performance of the proposed method on the NYUv2 dataset. We first trained and evaluated DCSN-EfficientNet and DCSN-RDN on the Middlebury dataset. Then, we evaluated the pretrained models on the Middlebury dataset, in order to study the generalization ability of the proposed method. After that, we fine-tuned both DCSN-EfficientNet and DCSN-RDN on the NYUv2 Depth dataset, in order to evaluate the performance of the DSR task on the data captured by depth sensors.

Finally, we evaluated the network trained for both depth completion and DSR tasks on the Matterport3D dataset. Note that, for the depth completion task, the network needs not only to fill in the missing depth values, but also to modify incorrect depth values in the raw depth map. Furthermore, the Middlebury and NyuV2 datasets do not have the ground truth for the depth completion task. To evaluate the DSR on the Middlebury and NyuV2 datasets, we downsampled the raw depth maps to generate the LR depth maps and kept the raw depth maps as the ground truth. On the other hand, the ground truth of the depth completion task is generated from the 3D reconstruction of the whole scene. Therefore, noise and misestimated depth values are also reduced in the ground truth for the depth completion task. Therefore, we did not test the DSR performance of the proposed network (which was trained on Matterport3D) on the Middlebury and NyuV2 datasets. Since the network is supposed to modify the depth map to reduce noise and false estimated depth values, the errors were higher than those of the model trained only for the DSR task.

### 4.1. Datasets

Middlebury stereo dataset [17,49,50,51,52]. The dataset we used consists of 59 RGB-D images from the Middlebury stereo dataset (7, 2, 9, 23, and 18 images from the 2001, 2003, 2005, 2006, and 2014 datasets, respectively). We trained our network with 55 images. Instead of using whole images to train the network, we randomly extracted small patches from the original images to be used as the input of the network. As the resolutions of the original depth maps vary, the sizes of the patches also vary, from 64×64 to 256×256. We also augmented the data by rotating and flipping the extracted patches. Finally, we evaluated the super-resolution performance of our proposed method on the *Cones, Teddy, Tsukuba, Venus, Jadeplant, Motorcycle, Playtable*, and *Flower* images (these images have been used in a previous work [9] on DSR)

NYUv2 dataset [53]. This dataset consists of 1449 pairs of aligned RGB and depth images from a variety of indoor scenes captured using a Microsoft Kinect. We used 795 samples to fine-tune the model, which was previously trained on the Middlebury stereo dataset, and 654 samples to test the performance of the proposed method. We split the dataset into training and testing sets, as in [53]. We did not conduct depth completion experiments on the NYUv2 depth dataset [53], as the ground truth for depth completion was not available.

Matterport3D [43]. This large-scale indoor dataset consists of over 110,000 RGB-D images of 90 scenes. For the depth completion task, Yinda et al. [13] generated the ground truth from reconstructed meshes for each scene in the Matterport3D dataset. Due to the large number of samples and the availability of the ground truth for depth completion, we decided to evaluate the proposed method on this dataset.

### 4.2. Evaluation Metrics

In this study, we used the following standard metrics to evaluate depth completion: RMSE, MAE, PSNR, and SSIM. Both RMSE and MAE directly evaluate the accuracy of the predicted depth maps; however, the RMSE is more sensitive to larger deviations, while the MAE is much less sensitive to the magnitude of the deviation. In other words, the RMSE can evaluate both the accuracy and consistency of the model (not producing outliers), while the MAE only focuses on the accuracy that the model can achieve overall. On the other hand, SSIM is used to evaluate the structural similarity between the ground truth and the completed depth map. We also evaluated the proposed method, based on the percentage of pixels within the error range *t*, as proposed in [13,30]. This assessment was used to further evaluate the number of inliers produced by the depth completion method. The error range is defined in Equation (6):(6)$$\delta_{t}=\frac{1}{w\times h}\sum_{i=0}^{w}\sum_{y=0}^{h},\delta_{t}\left(i,j\right)$$
where t∈1.05,1.10,1.25,1.252,1.253, *w* and *h* are the width and height of the HR depth map, respectively, and $\delta_{t}\left(i,j\right)$ is defined in Equation (7):(7)$$\delta_{t}\left(i,j\right)=\left\{\begin{array}{cc} 1, & max\left(\frac{D^{SR}\left(i,j\right)}{D^{HR}\left(i,j\right)},\frac{D^{HR}\left(i,j\right)}{D^{SR}\left(i,j\right)}\right)>t\\ 0, & \mathrm{otherwise} \end{array}\right..$$

### 4.3. Experimental Setup

In our work, we optimized the network parameters using AdamW [54], with the initial learning rate set to $10^{-4}$. The AdamW (or Adam with decoupled weight decay) algorithm is a variant of Adam [55], which is used to speed up convergence. Note that the initial learning rate is the upper bound of the learning rate for AdamW. The learning rates of the parameters are further modified by AdamW. We used the standard exponential decay rates and weight decay rate as follows: $\beta_{1}=0.9$, $\beta_{2}=0.999$, and λ=0.01. After every 10 epochs, we reduced the upper bound of the learning rate by half, in order to accelerate convergence. Finally, the model converged after 50 epochs. We implemented the depth completion model in Python 3.7, using the PyTorch 1.3 library. We conducted all the experiments on an Nvidia GeForce GTX 1080 Ti.

### 4.4. Comparison of Depth Completion Methods on the Matterport3D Dataset

In this experiment, we scaled the depth map and image to 320×256, as in [13,30]. The training process included two stages. In the first stage, we froze the Refinement Module and trained only the network for the DSR task. In the second stage, we trained the entire network, considering both the DSR and depth completion tasks. We used the PyTorch implementation of EfficientNet from the Segmentation Models library [56] in this work. Finally, we implemented only the Guidance branch with EfficientNet to limit the computational time and required memory. On a GeForce GTX 1080 Ti graphics card, the computational time of the network doubled when we implemented all branches with EfficientNet, but the quality of the results did not change significantly. Thus, we decided to implement only the Guidance branch with EfficientNet to gain speed.

Table 1 shows the comparison of the experimental results on the Matterport3D dataset, considering our method and the methods proposed by Zhang et al. [13] and Huang et al. [30]. Note that this is the result of the networks trained for depth completion only. The RMSE and MAE were calculated in meters. The results showed that the proposed method with the Guidance branch based on UNet was slightly worse than the method proposed in [30]. However, it is noticeable that it had a higher percentage of inliers, as shown by the δt and SSIM values. It seems to be that the UNet architecture slightly overfit the training data. Thus, UNet tended to produce more outliers on the test set. Therefore, it increased both the RMSE and MAE. On the other hand, EfficientNet has been shown to have better generalization on the ImageNet dataset. It can produce better results and outperform state-of-the-art methods. We provide a visual comparison of the depth completion task for the proposed method on the Matterport3D dataset in Figure 3.

### 4.5. Comparison of Depth Super-Resolution Methods on the Middlebury Dataset

In this section, we focus on evaluating the DSR performance of the proposed method on the Middlebury dataset. To improve the quality of the SR depth map, we also performed refinement based on the HR color image. First, we extracted the deep features from both the coarse SR depth map and the HR color image using the RDN [41]. Then, the features from the coarse SR depth map and the HR color image were concatenated. After concatenation, we applied another RDN to produce the refined SR depth map. Each RDN had eight convolutional layers with a sixty-four-channel feature map. Each convolutional layer in the RDN had 64 channels.

In this experiment, the upper bound of the learning rate was reduced by 10 percent every 25 epochs, due to the size of the dataset, instead of reducing the upper bound by half after every 10 epochs, due to the size of the dataset. The DSR network converged after 750 epochs.

We compared the upscaling results with integer scaling factors of ×2 and ×4 with the baseline upscaling methods, the methods proposed by Yang et al. [3], Kiechle et al. [4], Song et al. [9], and Truong et al. [40], and the proposed method, with respect to the RMSE; the results are provided in Table 2. In this experiment, the baseline upscaling methods were bicubic and Nearest Neighbor (NN). We highlight the best RMSE for each evaluation with bold text and the second best with underlines. In this work, we trained the network only with arbitrary scale factors between one and four, as the difference in the resolution of the depth map and the color image usually falls within this range. The proposed method obtained the best results in most cases and also had better performance on average. Two variants of the proposed method obtained very similar results, but DCSN-EfficientNet performed better when the scale factor was greater than ×2.5. Figure 5 shows the RMSE of different upsampling factors ($\left[1.1,1.2,1.3,\ldots ,3.7,3.8,3.9,4.0\right]$) with different methods. To obtain the upscaled depth maps for noninteger scale factors when using the method of Song et al. [9], we applied bicubic interpolation to the upscaled result at the nearest integer scale factor. For example, we downsampled the upscaled depth map with a scale factor of four, based on bicubic interpolation, to obtain the results for scale factor of three-point-seven. As mentioned in Section 3.1, although the interpolated results for noninteger scale factors of the method of Song et al. [9] were better than using the bicubic interpolation directly, the RMSE of [9] for the different scale factors on the Middlebury dataset was much higher than that of the proposed method, especially for larger scale factors. This could be due to the errors in the upscaled depth maps with integer scale factors. Thus, the second interpolation to rescale the upscaled depth map to the desired noninteger scaling factors may have amplified such errors. On the other hand, MetaDSR [40] had a very similar RMSE to the proposed method for scale factors in the range [2.5,4.0]. However, the proposed method had much better performance for smaller scale factors. We believe that the main factor affecting the results for smaller scale factors was the different strategy used to project the feature maps onto the SR image plane. In [40], the feature map extracted from the LR depth map was projected to create an SR feature map that had the same spatial dimension (resolution) as the SR depth maps. Then, the depth value at location (i,j) in the SR depth map was produced by applying a 3×3 kernel, generated by the meta-upscale module at (i,j). In [40], this reduced the block effect on the final SR depth map. However, it also reduced the sharpness of the edges on the SR depth maps, as the meta-upscale module does not consider edge information (only scale and relative location). In addition, as scale factors in the range (1,2.5] are simpler than the larger scale factors, the gaps in RMSE between the proposed method and MetaDSR [40] were much higher for lower scale factors. We show a visual comparison of the proposed method on the noise-free Middlebury dataset with different scale factors in Figure 6.

We also tested the proposed method on the same setup with noisy input, as in [9]. The proposed method produced the best average RMSE on the noisy Middlebury dataset, as shown in Figure 7. We illustrate the visual results of the proposed method for DSR on the noisy Middlebury dataset in Figure 8.

We further assessed the ability of the proposed method to handle noisy data with missing depth values [58]. This dataset was generated based on the Middlebury dataset, by removing the depth values based on two types of masks: random missing masks and textual masks. Although the distribution of missing depth values in this dataset was not realistic, it was still useful to evaluate the performance of the proposed method in handling noisy data. In this experiment, we computed the PSNR value of the proposed method, as this was the main evaluation metric for this dataset. As this dataset was mainly designed for the noise reduction task, we computed the denoised depth map by applying the proposed method to the depth maps with the scale factor of 1.0. Table 3 shows that our method achieved the best average PSNR value on this dataset, compared to the state-of-the-art methods. This proves that the proposed method can handle different types of noisy inputs better than the state-of-the-art methods. Note that the DCSN-EfficientNet variant of the proposed method achieved much better results than DCSN-RDN, as it included a Depth Completion Module. Thus, DCSN-EfficientNet could handle missing depth values much better than DCSN-RDN. Figure 9 shows the average PSNR values obtained by the proposed method on the missing depth value dataset [58] for the DSR task. We show illustrations of DSR for the proposed method on depth maps with missing values in Figure 10.

Finally, we compared the computational time for upscaling the Art depth map to its full resolution (1390×1110), between the proposed method and the other considered methods, in Figure 11. In [9], the author consecutively applied the base network for a scale factor of two to upscale the depth maps to times four. Thus, the computational time of the method in [9] sharply jumps up at the scale factor of three, as we interpolated the SR depth maps from the times four upscaled depth map. In contrast, the DCSN-RDN variant of the proposed method still managed the same computation time at all scaling factors. DCSN-EfficientNet, with a much larger network, acquired 15 FPS; which is still suitable for real-time applications.

### 4.6. Comparison of Depth Super-Resolution Methods on the NyuV2 Dataset

In this experiment, we further evaluated the accuracy of different DSR methods and the two above-mentioned variants of the proposed method on the NYUv2 dataset. All methods were trained on the Middlebury stereo dataset, without any fine-tuning on the NYUv2 dataset. Figure 12 shows the RMSE (in meters) of the different methods for different upsampling factors $\left[1.1,1.2,1.3,\ldots ,3.9,4.0\right]$. The proposed method was the only method that had comparable results to bicubic interpolation. In fact, compared to the bicubic interpolation, the proposed method had a better RMSE in 66% of the cases. This was expected, as the characteristics of the disparity maps in Middlebury were very different from the depth maps captured by depth sensors. For example, the depth maps in NyuV2 (Figure 13) contained many noisy pixels, while the depth maps in the Middlebury dataset were noise-free. The results showed that the proposed method based on RDN had the best generalization, compared to state-of-the-art methods. On the other hand, the DCSN-EfficientNet variant had slightly worse results, as it was much larger than DCSN-RDN. Figure 14 shows the RMSE of the different methods for different upsampling factors after fine-tuning on the training set of the NYUv2 dataset. The proposed method had the best performance, with regard to the RMSE, in most cases. We show examples of DSR on NYUv2 in Figure 13.

### 4.7. The Performance for Both the DSR and Depth Completion Tasks

In this experiment, we evaluated the performance of both DSR and depth completion. As the resolution of 320×256 is quite small, the downscaled depth map could lead to the loss of many details. Thus, we rescaled the depth maps in the Matterport3D dataset to different resolutions. We evaluated the results on these resolutions for the HR depth maps. The resolutions included 320×256, 640×512, and 1280×1024.

Figure 15 shows the results for the RMSE of the proposed network with the Guidance branch based on EfficientNet for depth completion and DSR on the Matterport3D dataset. For the 1240×1024 resolution, we first reduced the resolution of the input images and depth maps by half. Then, we leveraged the DSR module of the proposed method to upscale back to the 1240×1024 resolution. This helped to reduce the required memory and computational time of the proposed method. Furthermore, the receptive field of the proposed method was also not able to cover the whole area of large missing regions. The errors of the proposed method at the resolution of 320×256 tended to grow larger when the scale factor was increased. As mentioned above, this was predictable, due to the low resolution of the input. For the resolutions of 640×512 and 1240×1024, the results were worse with the scaling factor of 1.0. However, the RMSE of the upscaled depth maps slightly decreased when the scale factor increased. This was because the size of the holes in LR depth maps are much larger when the scale factors are smaller. Thus, it is much harder to fill in the depth information. Although the task is much more complicated, the proposed method still had errors that were not too different than when just performing depth completion. We show a visual comparison of DSR and depth completion by the proposed method on the Matterport3D dataset with different resolutions and scale factors in Figure 16.

Table 4 shows a comparison of the computational time for the depth completion task (at 320×256 resolution). The methods in [13] and [30] used the same architecture for surface normal estimation and boundary estimation. Thus, we also included the computational time of this step, as it is required for these methods to fill in missing depth values. We used the official source code of these methods to measure the computational time. The computational time required by the proposed method was much lower than that of the state-of-the-art methods. As the global optimization in [13] needs to be performed on a CPU, the computational time of this method was the highest. On the other hand, Huang et al. [30] proposed a method that uses multiple DCNNs for different purposes, such as surface normal estimation, coarse depth completion, and refinement. The proposed method has multiple stages, but each stage contains only a few convolutional layers. Thus, the proposed method succeeded in keeping the computational time low. We also report the FPS of the proposed method on the 1240×1024 resolution images in Figure 15. Note that we downscaled the depth maps and color images by the scaling factor *r* and used the network to upscale the LR depth map back to the original resolution. The results were measured using the average FPS of the proposed method on the entire Matterport3D dataset. This provided us with a closer look at the speed of the proposed network when performing depth completion and DSR in real applications.

## 5. Conclusions

In this paper, we proposed a novel DSR method that simultaneously performs DSR and depth completion for arbitrary scale factors. The proposed method only requires storing a single model to perform the DSR task for multiple arbitrary scale factors and outperformed state-of-the-art DSR and depth completion methods. When combining DSR and depth completion, the proposed method achieved excellent results, compared to the other methods, which only consider depth completion. Finally, by leveraging the features extracted from the depth completion module for DSR, the proposed method also requires a relatively low computational time. Furthermore, the DSR network was designed to reuse the extracted features from the depth completion network on the LR map. Thus, the proposed method can minimize both the size of the depth completion network and the computational time of the depth completion network.

Although the proposed model applies spatial dimension reduction for feature extraction, the depth details of large empty regions are often blurred. This happens because very little depth information is transmitted to the pixels in the center of the hole region. Therefore, we will try to develop an iterative model to solve this problem in future research. The depth information can then be transmitted more efficiently to the deeper regions of the void. Although the proposed model can be extended to videos by processing each frame independently, the lack of temporal features may negatively affect the final result of the model. In the future, we will focus on modifying the model to allow for the extraction of temporal features from video sequences for depth completion and DSR.

## Figures and Tables

**Figure 1 sensors-21-04892-f001:**
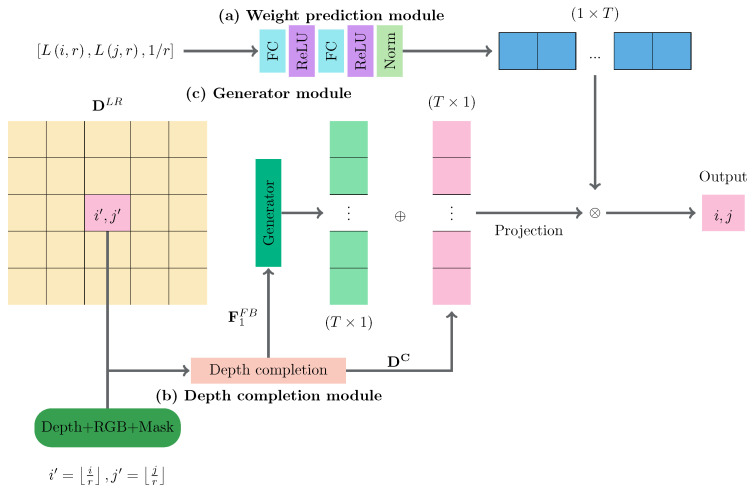
The architecture of the proposed networks for depth completion and depth super-resolution with arbitrary scale factors. This figure depicts the process used to synthesize the output at location i,j in the SR depth map: (**a**) the Weight Prediction Module; (**b**) the Depth Completion Module; and (**c**) the Generator Module. Note that ⊕ and ⊗ represent the elementwise summation of two matrices and matrix multiplication, respectively.

**Figure 2 sensors-21-04892-f002:**
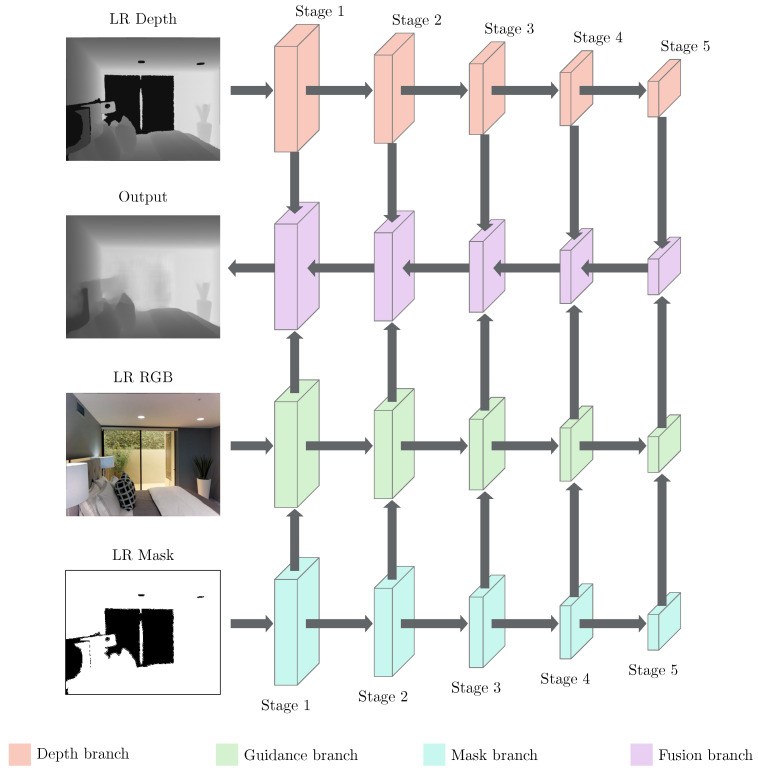
Architecture of the Depth Completion Module. The module consists of four DCNN branches: A Depth branch, a Guidance branch, a Mask branch, and a Fusion branch.

**Figure 3 sensors-21-04892-f003:**
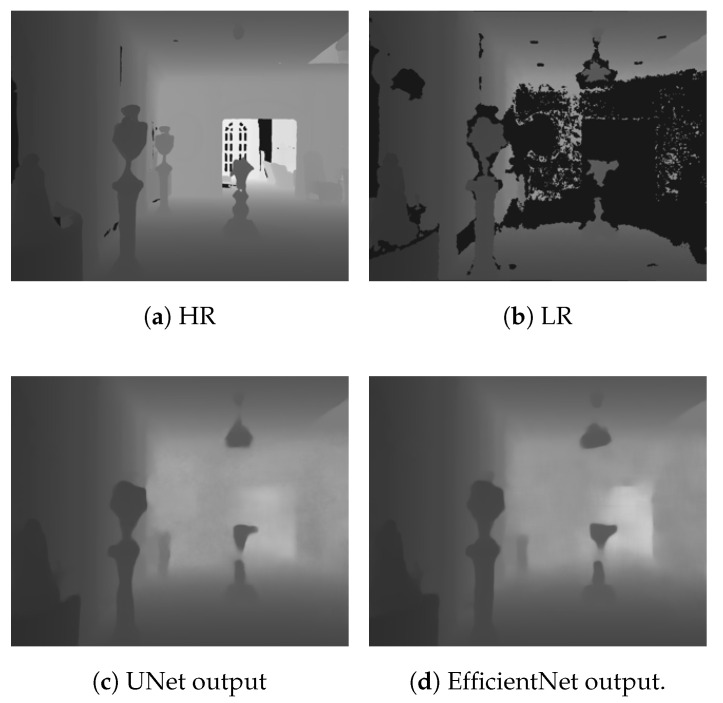
The visual comparison of the depth completion on the Matterport3D dataset [43]. Note that (**c**) was produced using the Guidance branch based on UNet [26] and (**d**) was produced using the Guidance branch based on EfficientNet [44].

**Figure 4 sensors-21-04892-f004:**
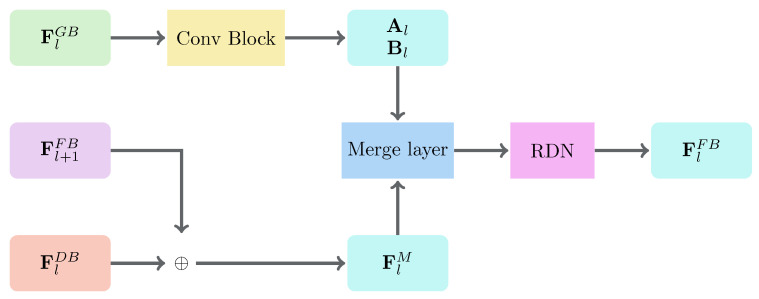
The architecture of the Fusion Block. Note that ⊕ represents the elementwise summation of two matrices.

**Figure 5 sensors-21-04892-f005:**
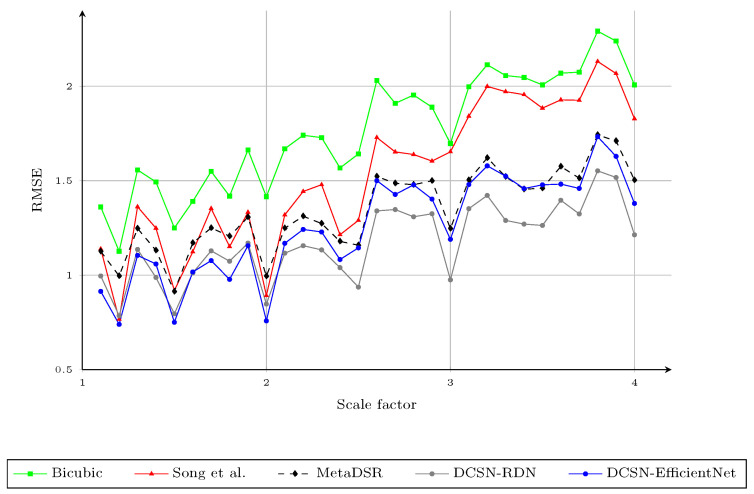
The RMSE of different upsampling factors $\left[1.1,1.2,1.3,\ldots ,3.8,3.9,4.0\right]$ using different methods. We report the results of the modality in [40] trained with arbitrary scale factors, instead of that trained with specific scale factors. Note that the results for the method of Song et al., in noninteger cases, were obtained by applying bicubic interpolation to the upscaled results of the nearest integer scale factor.

**Figure 6 sensors-21-04892-f006:**
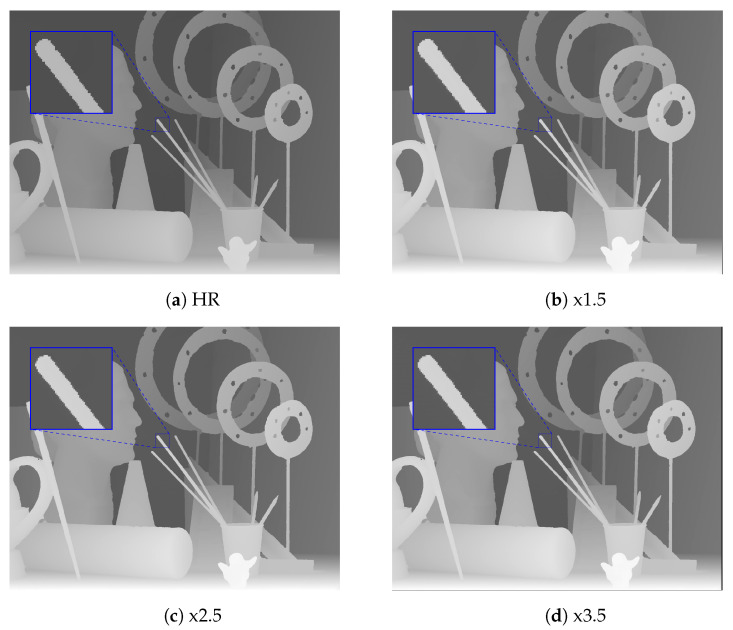
Visual comparison of DSR for the proposed method on the Art depth map with different scale factors.

**Figure 7 sensors-21-04892-f007:**
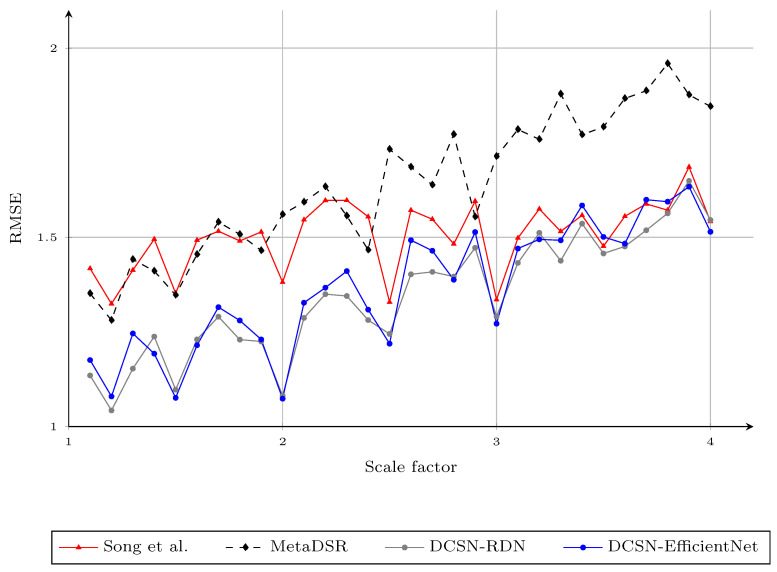
The RMSE of different methods for different upsampling factors $\left[1.1,1.2,\ldots ,3.9,4.0\right]$ on noisy input depth maps. We report the results of the model in [40] trained with arbitrary scale factors, rather than the model trained with specific scaling factors, for a fair comparison. Note that the results of the method of Song et al. for noninteger scale factors were obtained by applying bicubic interpolation to the upscaled results of the nearest integer scale factor.

**Figure 8 sensors-21-04892-f008:**
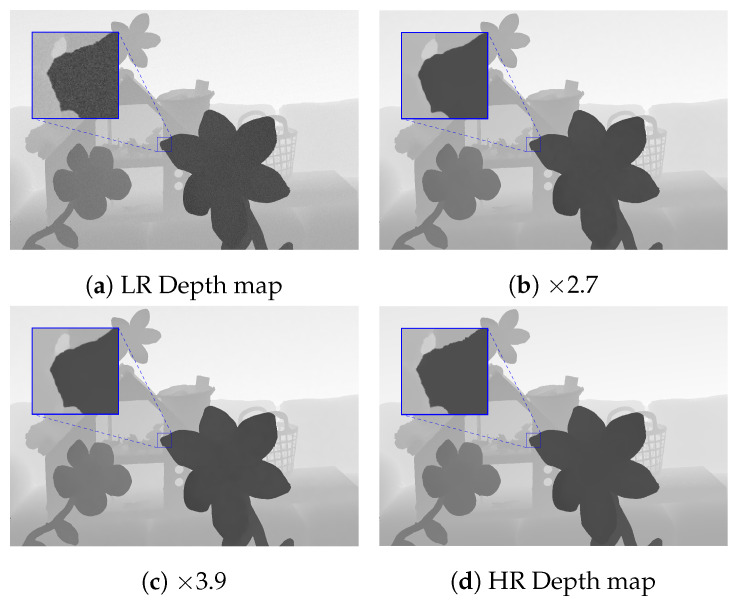
Visual comparison of DSR by the proposed method on the noisy Middlebury dataset [17,49,50,51,52] with different scale factors.

**Figure 9 sensors-21-04892-f009:**
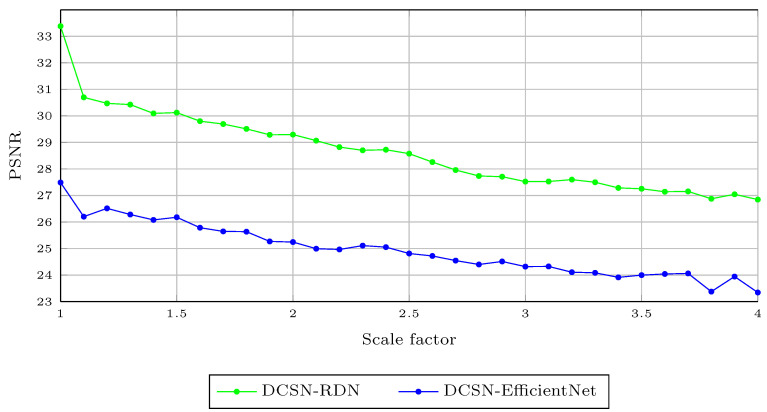
Average PSNR for different upsampling factors $\left[1.1,1.2,1.3,\ldots ,3.7,3.8,3.9,4.0\right]$ using the proposed method on the missing depth value dataset [58].

**Figure 10 sensors-21-04892-f010:**
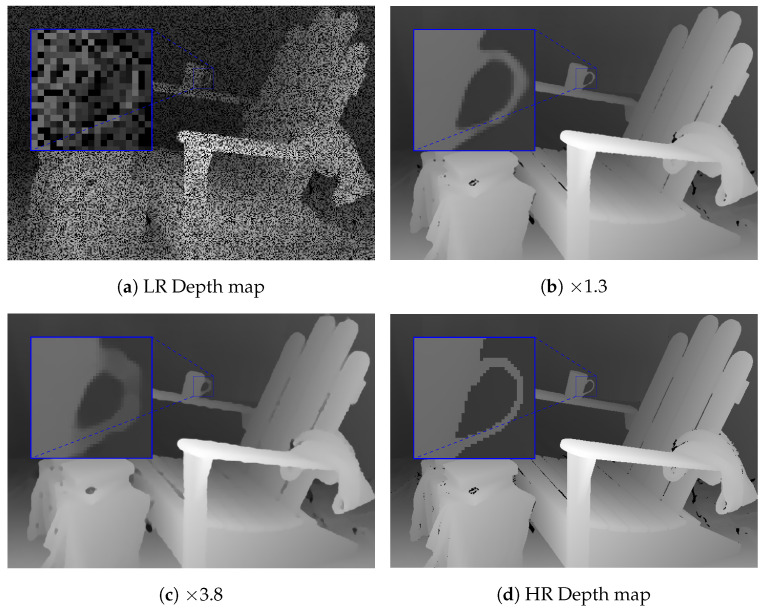
Visual comparison of DSR on the noisy Middlebury dataset [17,49,50,51,52] with different scale factors.

**Figure 11 sensors-21-04892-f011:**
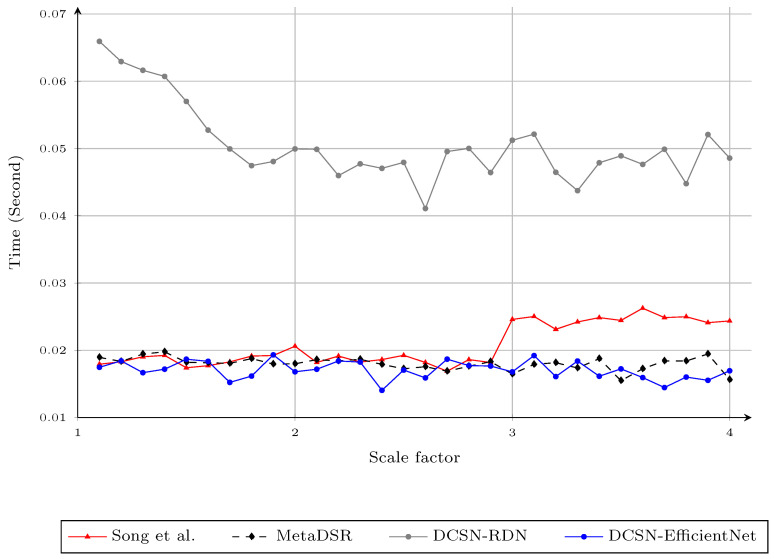
Computational time (seconds) for different upsampling factors $\left[1.1,1.2,\ldots ,3.9,4.0\right]$ using different methods on the Art depth map. Note that the results for the method of Song et al. for noninteger scale factors were obtained by applying bicubic interpolation to the upscaled results of the nearest integer scale factor.

**Figure 12 sensors-21-04892-f012:**
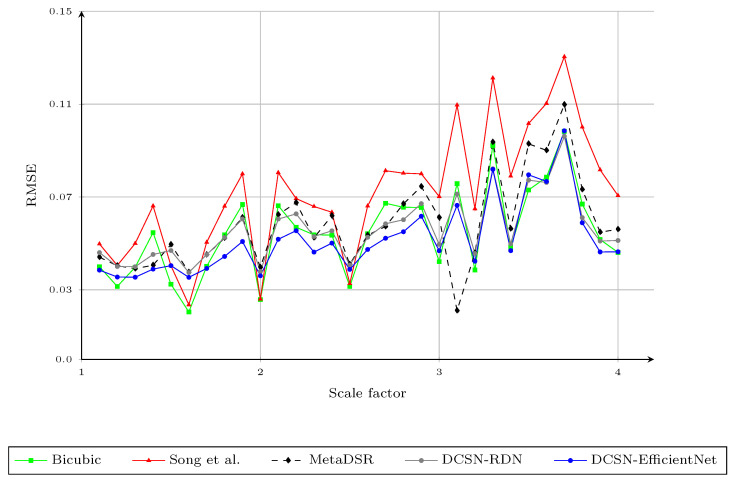
The RMSE of different methods for different upsampling factors $\left[1.1,1.2,\ldots ,3.9,4.0\right]$ on the NYUv2 dataset without fine-tuning. We report the results of the modality in [40] trained with arbitrary scale factors, instead of the one that was trained with specific scale factors. Note that the results of the method of Song et al. for noninteger scale factors were obtained by applying bicubic interpolation on the upscaled results of the nearest integer scale factor.

**Figure 13 sensors-21-04892-f013:**
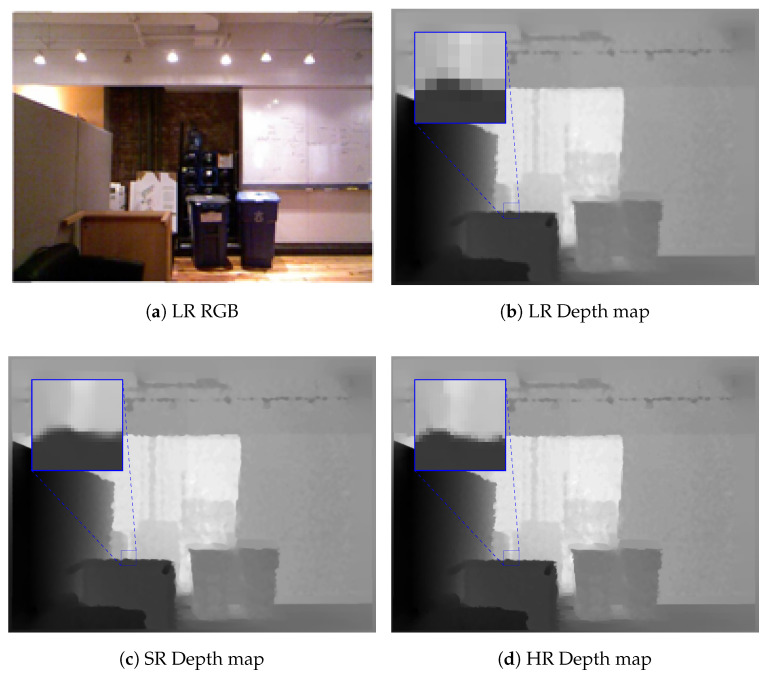
Visual comparison of the DSR on the NYUv2 dataset [53] with a scale factor of 3.3.

**Figure 14 sensors-21-04892-f014:**
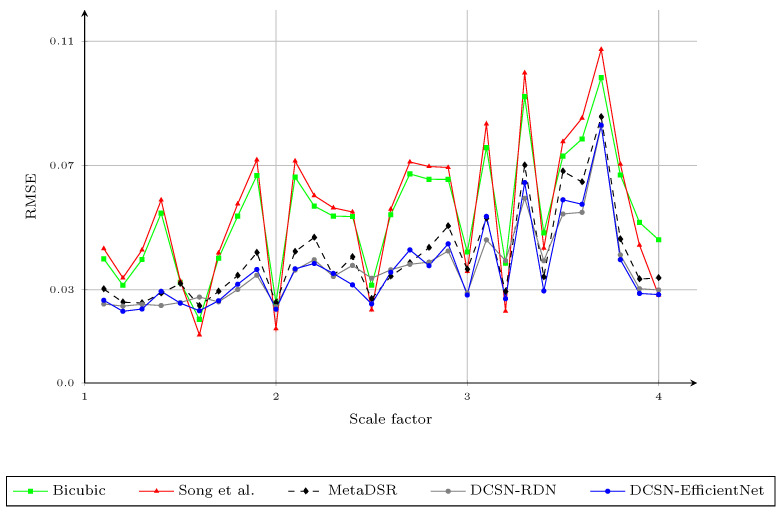
The RMSE of different methods for different upsampling factors $\left[1.1,1.2,\ldots ,3.9,4.0\right]$ on the NYUv2 dataset after fine-tuning. We report the results of the modality in [40] trained with arbitrary scale factors, instead of the one that was trained with specific scale factors. Note that the results of the method of Song et al. for noninteger scale factors were obtained by applying bicubic interpolation on the upscaled results of the nearest integer scale factor.

**Figure 15 sensors-21-04892-f015:**
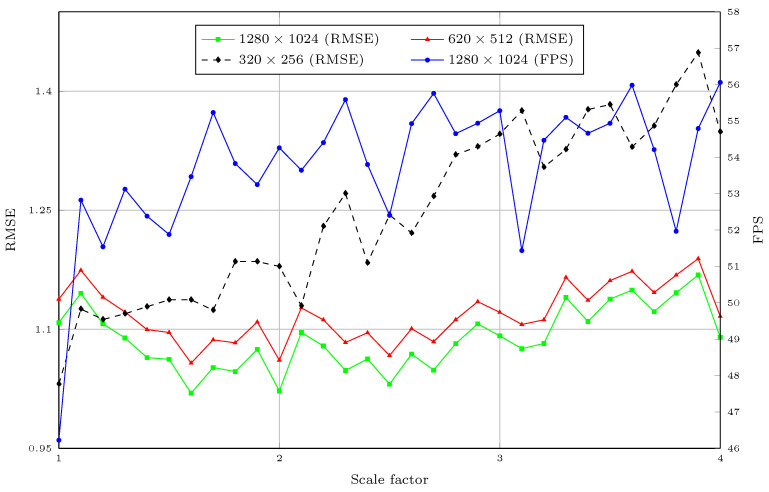
The RMSE and FPS of different upsampling factors $\left[1.0,1.1,1.2,1.3,\ldots ,3.7,3.8,3.9,4.0\right]$ of the proposed method on the Matterport3D dataset.

**Figure 16 sensors-21-04892-f016:**
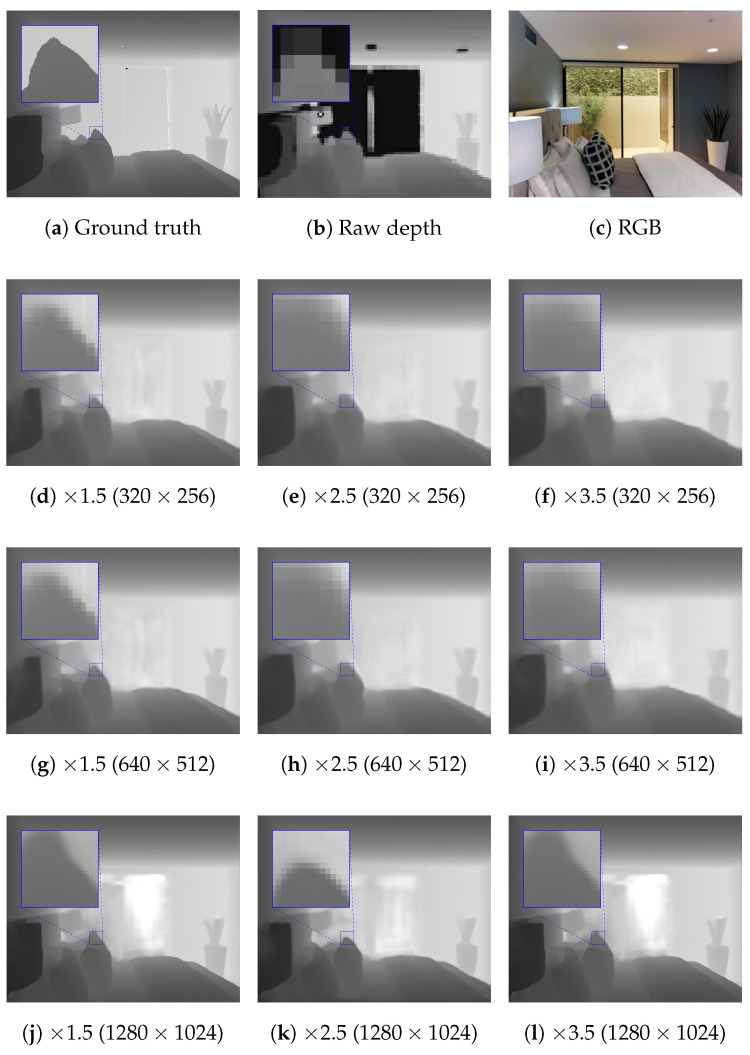
Visual comparison of DSR and depth completion on the Matterport3D dataset [43] using the resolutions of 320 × 256, 640 × 512, and 1280 × 1024.

**Table 1 sensors-21-04892-t001:** The RMSE, MAE, SSIM, and the percentages of pixels within different error ranges for depth completion on the Matterport dataset. The best result is shown in bold. The RMSE of the FCN is from [21], that for the MRF is from [57], and that for the bilateral filter was given in [30]. For the RMSE and MAE, lower is better. For SSIM and the error ranges, higher is better.

	RMSE	MAE	SSIM	$\delta_{1.05}$	$\delta_{1.10}$	$\delta_{1.25}$	$\delta_{1.25^{2}}$	$\delta_{1.25^{3}}$
Bilateral	3.0186	3.4961	0.507	0.385	0.497	0.613	0.689	0.730
MRF [57]	3.6726	4.2625	0.692	0.506	0.556	0.651	0.780	0.856
FCN [21]	3.6726	4.2625	0.605	0.397	0.527	0.681	0.808	0.868
Zhang et al. [13]	1.316	0.461	0.762	0.657	0.708	0.781	0.851	0.888
Huang et al. [30]	1.092	0.342	0.799	0.661	0.750	0.850	0.911	0.936
DCSN-UNet	1.104	0.338	0.800	0.702	0.776	0.860	0.913	0.937
DCSN-RGBM	1.045	0.308	0.809	0.712	0.788	0.870	**0.922**	**0.945**
DCSN-EfficientNet	**1.031**	**0.305**	**0.811**	**0.717**	**0.794**	**0.871**	**0.922**	**0.945**

**Table 2 sensors-21-04892-t002:** The RMSE of upsampling factors (×2.0 and ×4.0) for different methods. We report the results of the modality in [40] trained with arbitrary scale factors, instead of that trained with specific scale factors. The best result is shown in bold, and the second best is underlined. For other methods, we used the RMSE given in [40] (for which lower is better).

Method	×2.0				×4.0			
	Venus	Tsukuba	Cones	Teddy	Venus	Tsukuba	Cones	Teddy
Bicubic	1.058	2.582	2.666	2.601	1.532	3.701	3.620	3.358
NN	0.997	2.397	2.440	2.448	1.525	3.450	3.439	3.282
Yang et al. [3]	1.237	5.631	2.421	1.894	2.755	13.174	5.139	4.066
JID [4]	0.688	3.742	1.745	1.268	0.963	5.903	3.037	1.804
Song et al. [9]	**0.278**	1.651	0.876	0.761	0.499	4.384	2.196	1.517
MetaDSR [40]	0.484	1.413	1.040	1.042	0.697	2.487	1.393	1.434
Ours	0.420	**1.152**	**0.651**	**0.567**	**0.628**	**1.962**	**1.220**	**1.201**

**Table 3 sensors-21-04892-t003:** Average PSNR of different methods on the missing depth value dataset [58] for the denoising task (a higher PSNR is better). LRL0 denotes Low Rank Matrix Completion and LRL0${}^{\psi }$ denotes Low Rank Matrix Completion with low gradient regularization [58].

LRL0 [58]	LRL0${}^{\psi }$ [58]	WNNM [59]	DCSN-RDN	DCSN-EfficientNet
22.625	23.029	27.375	33.383	27.489

**Table 4 sensors-21-04892-t004:** The computational time (seconds) of the depth completion on the Matterport dataset. Note that the computational times of both Zhang et al. and Huang et al. were reported as the computational time of surface normal estimation and boundary estimation. Both of the above methods require the surface normals and boundaries to be extracted from the RGB image for depth inference. This step took 0.02 s (Huang et al. used the same method as Zhang et al. to estimate the surface normals and boundaries).

Method	Surface Normal and Boundary Estimation (s)	Depth Inference (s)	Total (s)
Zhang et al. [13]	0.020	1.430	1.450
Huang et al. [30]	0.020	0.005	0.025
Ours	N/A	0.008	0.008

## Data Availability

The data presented in this study are available on request from the corresponding author.

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
