# Peer review of "Depth Completion and Super-Resolution with Arbitrary Scale Factors for Indoor Scenesâ€"

_sensors, 2021, doi:10.3390/s21144892_

Round 1
Reviewer 1 Report
In this article, the authors propose a method for dealing with depth maps inpainting and DSR (with arbitrary scale factor) problems.
The article is well written, the underlying problem is clearly presented, and the state of the art is deeply described. The method is adequately illustrated in justified steps. The result evaluation is sound in the chosen metrics, in the chosen test datasets, and the comparison with state of the art. The discussion is extensive; perhaps greater conciseness of this section would benefit the reader.
On the contrary, the conclusions are essential; please summarize the major observations highlighted by the experiments.
Line 217. Do you mean Fusion Branch?
The authors have inserted many comparative images, for the most part poorly described and not referred to within the text. Figure 4, what should the reader pay attention to? Add a comment. Figures 6,7,9: the article is quite long, is there any way to visually resume them? These images also lack commentary and reference in the text.
Can all RMSE charts be grouped into a single figure?
Author Response
Dear Sir or Madam,
Thank you so much for your comments. I have tried to fixed the problems which you have pointed out. Please find the details of the response in the attached file. Yours sincerely, Anh Minh Truong.

Reviewer 2 Report
“Depth completion and depth super-resolution with arbitrary scale factors for indoor scenes” by Anh Minh Truong, Wilfried Philips, and Peter Veelaert
Summary:
This study proposes a new deep learning method for upscaling depth maps. The method allows for depth completion (filling in missing or erroneous depth values) and depth super-resolution (upscaling the depth image for a camera to match the resolution of a corresponding RGB image) that can handle arbitrary scale factors. One piece of novelty is that the proposed solution can be used for either DSR or Depth completion (or both) rather than just for one of the two tasks. The method was tested on 3 datasets involving several experiments.
Review:
Overall, the manuscript has some very interesting ideas and approaches. However, the writing is frequently unclear. It frequently provides extra unnecessary information and needs to be condensed. Many of the figures need to be moved much closer to the text that references them. Many figures are included that are not referenced in the text (I can only find references for Figures 1, 2, 3, 8, 10, 11, 12, 14, 15, 16, 17, 18, and 19 – meaning that at least 6 figures are not referenced in the paper). Overall, the disorganization, lack of detail/clarity, and addition of wordy, unneeded information detract significantly from the information provided in the manuscript. Significant reorganization, revision, and editing are required. Several examples of these issues are described below.
Grammatical mistakes are littered throughout the article, beginning in the first sentence of the introduction (have instead of has in line 17). Some other examples:
- Line 23: No comma after “Even though”.
- Sentence beginning on Line 40: This sentence does not make sense. It needs to be rewritten.
- Redundant sentences repeating the same information: Beginning on line 42: “In this pipeline, the depth completion is applied to the result of the DSR” and beginning on line 44: “Meanwhile, their method just applied the depth completion method on the super-resolution (SR) depth maps.”
- Line 48: “Thus, the size of the depth completion network…” – missing “the”
- Line 55: “depending” rather than “depends”
- Line 64: “of a much” -> “of much”. Remove the “a”
Rather than continuing to find other grammatical mistakes, I suggest that the authors take another very close inspection of grammar throughout the submitted manuscript. The paragraphs starting on Line 121 and line 134, for example, have several errors.
Many embedded links (DSR, LR, and SR) do not work properly and would be better if they either worked properly or simply were not links. (Another quick grammatical issue: Line 63 cannot just end with “LR”. It could be written as a “low resolution images” or “LR images” but not just “LR”)
Line 57: “This happens quite a lot on the raw depth maps.” – how much is quite a lot? Please provide a number or reference for this claim. (EX: if you used a RealSense R200, a quick set of calculations could provide you with a specific frequency with which data is missing, and you could make a claim that the RealSense R200 averages X missing pixels per frame)
Line 68: This bullet point is not part of the novel approach or needs to be rewritten. It is a result, so it should be reported in another location (not in a bulleted list pointing out the proposed novel approach). It is also extremely long and includes multiple points. If this is a bulleted list, the points should be individual, clear, and succinct.
Lines 96-100 seem to contradict each other. It is claimed that “accurately estimating the depth from just a single RGB image remains extremely challenging” while also claiming that “many studies have shown promising results in recent years”. This needs to be clarified.
Line 101: Do not start a sentence with a reference. One suggested fix: “Eigen et al. estimated the depth map… “ with a reference at the end of the sentence.
Line 118: This is the first mention of the importance of “real-time” applications in your solution. This should be discussed earlier in the manuscript, since it highlights the importance of faster processing (EX: using a low resolution image for completion before upscaling).
Figures are frequently far too separated from their references. EX: Figure 1 is on page 2, while the reference to Figure 1 is on page 7. Figures should be close to where they are referenced.
Figure 1 contains far too much information in the caption. The details of the architecture should be discussed in the paper, using Figure 1 as a reference. The caption should not repeat all the details. In fact, captions for most tables and figures need to be reduced.
Figure 2 – minor point, but shouldn’t the output be at the bottom so that everything flows to the output?
Line 260 – “… greatly reduced.” – by how much? Is there a reference here? Provide specific values! This issue comes up frequently. EX: Line 263 “… very small …” how small? Provide specific values!
Line 316 – why these specific images? Were some left out? If so, why?
Line 322 – if the model if fine tuned on NYUv2, then is the new “fine-tuned” model re-applied to the Middlebury dataset? If not, then two different models are being tested here. Please be clear about which model(s) are applied to which dataset(s). If different versions are used, please indicate why.
The methods and results are difficult to follow and should be condensed and clarified. EX: dataset descriptions and evaluation metrics should be provided in the methods, not results.
Line 336 – 338: This is unclear. EX “we optimize the network parameters by Adam” – does this mean using Adam? Or are you using AdamW? The grammatical and clarity errors here make this section too difficult to understand clearly.
Additional issues exist, but the disorganization and lack of clarity make these difficult to identify.
Author Response

(The authors gave the same response as above.)

Round 2
Reviewer 2 Report
“Depth completion and depth super-resolution with arbitrary scale factors for indoor scenes” by Anh Minh Truong, Wilfried Philips, and Peter Veelaert
Summary:
This study proposes a new deep learning method for upscaling depth maps. The method allows for depth completion (filling in missing or erroneous depth values) and depth super-resolution (upscaling the depth image for a camera to match the resolution of a corresponding RGB image) that can handle arbitrary scale factors. One piece of novelty is that the proposed solution can be used for either DSR or Depth completion (or both) rather than just for one of the two tasks. The method was tested on 3 datasets involving several experiments.
Review:
The authors have corrected many of the specific issues provided, and the paper is significantly better than the original submission. However, as noted in the prior review, there are still many more grammatical errors, and there are some organizational problems.
Now that I understand more of the details due to the improved language, grammar, and organization, my largest concern is that several different models are run and tested on the three datasets, and it appears that the best results from each variant are reported. A pretrained model was tested on the Middlebury dataset, a fine-turned model on NyuV2, and three variants on the Matterport3D. However, for clear results, the same model should be evaluated on all 3 datasets. How can you guarantee any version of the model is not overfitted for a particular dataset? For example, in Reference 9 (paper by Song et. al.), section 4.7, there is an analysis of generalizability that includes retraining, and most importantly, evaluation of one network on multiple datasets.
Additional minor notes:
Table 3 does not fit on the page (perhaps just because of the track changes?)
Table 4 should list N/A rather than 0.000 for your method’s surface normal and boundary estimation for clarity.
The limitations of your method should be described in the conclusion.
There are still some grammatical and spelling issues, even in the updated text. The author has indicated that they have limitations with English, and I recommend that they take advantage of services for correcting English or try to appeal to a native English-speaking colleague to assist. Even some built in spelling correction tools would be useful for some of the mistakes seen.
Three examples found at a quick glance:
Line 174: “inpainting” rather than “inpaiting” (misspelling).
Line 252: “The inpainted depth map then projected onto the SR image plane.” (Missing “is” between “map” and “then”)
Line 329: “However, it might have bad influence…” -> “However, it might have a bad influence…”
The conclusion needs a thorough grammar check.
Author Response
Dear Sir or Madam,
Thank you so much for your comments and suggestions. We have tried to resolve the problems you have mentioned in the review report. You can find our responses in the attachment. We also found a bug in the latex track changes which does not show the citations appropriately. We cannot find the way to work around. Thus, we also attach a copy of the manuscript.
Thank you so much for your help.
Yours sincerely,
Anh Minh Truong.

Round 3
Reviewer 2 Report
“Depth completion and depth super-resolution with arbitrary scale factors for indoor scenes” by Anh Minh Truong, Wilfried Philips, and Peter Veelaert
Summary:
This study proposes a new deep learning method for upscaling depth maps. The method allows for depth completion (filling in missing or erroneous depth values) and depth super-resolution (upscaling the depth image for a camera to match the resolution of a corresponding RGB image) that can handle arbitrary scale factors. One piece of novelty is that the proposed solution can be used for either DSR or Depth completion (or both) rather than just for one of the two tasks. The method was tested on 3 datasets involving several experiments.
Review:
Thank you for clarifying the evaluation process. This addressed the most significant concern from my prior review. The grammatical corrections made throughout the paper were very helpful. However, I am still confused by the evaluation process and have some additional concerns.
1) The corrections again introduce grammatical errors. See, for example, “we select Middlebury Stereo…” – line 312 – and “this dataset is not capture by depth sensors.” – Line 314. Again, I strong urge the authors to have revisions reviewed for spelling and grammatical mistakes by a native English speaker.
2) In my prior review, request #4 was “The limitations of your method should be described in the conclusion.” The authors responded with: “Response: Thank you so much for your comment. I have added the discussion of the proposed method to the discussion section.”
However, there is no discussion section, and I could not identify any area in which the limitations were discussed.
3) Please respond to the points below to ensure the evaluation process is clear:
a) You trained your network for DSR and depth completion tasks separately on which data/dataset?
b) You evaluate the DSR performance on the Middlebury and NYUv2 datasets. You do this by evaluating the pretrained dataset from step (a) on Middlebury.
c) You also evaluate a version of your pretrained network that has a depth completion module on the Middlebury dataset? (Lines 454-457 indicate this, but it is unclear in the earlier discussion from lines 312-320)
d) You fine tune your DSR network on the NYUv2 dataset because it was captured by a depth sensor?
e) You train and evaluate your DSR network from step (a) on the Matterport 3D dataset?
f) You also train and evaluate a depth completion network (also from step (a)) on the Matterport3D dataset?
g) Ultimately steps (e) and (f) are combined to provide your final proposed model?
h) The model from step (g) cannot be evaluated on Middlebury or NyuV2 because the combination of DSR and depth completion fixes errors that are not corrected for in these two datasets. In other words, DSR and depth completion are now simultaneously performed by the model from step (g), and they cannot be evaluated separately any longer?
Regardless of the responses above, following the discussion in the paragraphs from lines 308-330 is very difficult. It is also not clear which versions of the model are trained on which datasets. The term “our” model is used frequently, although there appear to be at least 3 models. Overall, this language needs to be updated so the reader can have a clear understanding of how many models are evaluated, which data they are trained on, which data they are tested on, and why each of these decisions are made.
Author Response
To Whom It May Concern,
Thank you so much for your comments. We have submitted the manuscript to the English editing service of Sensors. We also have modified the manuscript as suggested in the comments. Please find the detailed answers in the attachment.
Yours sincerely,
Anh Minh Truong.
